# A Schematic Approach to Defining the Prevalence of COL VI Variants in Five Years of Next-Generation Sequencing

**DOI:** 10.3390/ijms232314567

**Published:** 2022-11-23

**Authors:** Gemma Marinella, Guja Astrea, Bianca Buchignani, Denise Cassandrini, Stefano Doccini, Massimiliano Filosto, Daniele Galatolo, Salvatore Gallone, Fabio Giannini, Diego Lopergolo, Maria Antonietta Maioli, Francesca Magri, Alessandro Malandrini, Paola Mandich, Francesco Mari, Roberto Massa, Sabrina Mata, Federico Melani, Maurizio Moggio, Tiziana E. Mongini, Rosa Pasquariello, Elena Pegoraro, Federica Ricci, Giulia Ricci, Carmelo Rodolico, Anna Rubegni, Gabriele Siciliano, Martina Sperti, Chiara Ticci, Paola Tonin, Filippo M. Santorelli, Roberta Battini

**Affiliations:** 1Department of Neuroscience, IRCCS Stella Maris Foundation, 56128 Pisa, Italy; 2Department of Clinical and Experimental Medicine, University of Pisa, 56126 Pisa, Italy; 3Department of Clinical and Experimental Sciences, University of Brescia, 25123 Brescia, Italy; 4NeMo-Brescia Clinical Center for Neuromuscular Diseases, 25064 Gussago, Italy; 5Neuromuscular Center, AOU Città della Salute e della Scienza, University of Torino, 10124 Torino, Italy; 6Department of Medical, Surgical and Neurological Sciences, University of Siena, 53100 Siena, Italy; 7ASL8, Center Sclerosi Multipla, 09134 Cagliari, Italy; 8Neurology Unit, IRCCS Fondazione Cà Granda Ospedale Maggiore Policlinico, 55031 Milan, Italy; 9IRCCS Ospedale Policlinico San Martino, 16132 Genoa, Italy; 10Meyer Children’s Hospital, University of Florence, 50121 Florence, Italy; 11Department of Systems Medicine, Tor Vergata University of Rome, 00133 Rome, Italy; 12Department of Neuromusculoskeletal and Sense Organs, SOD Neurology 1, Azienda Ospedaliero-Universitaria di Careggi, 50134 Florence, Italy; 13Neuromuscular and Rare Disease Unit, IRCCS Fondazione Cà Granda Ospedale Maggiore Policlinico, 20122 Milan, Italy; 14Department of Neurosciences “Rita Levi Montalcini”, University of Turin, 10124 Turin, Italy; 15Department of Neurosciences, University of Padua, 35122 Padova, Italy; 16Department of Clinical and Experimental Medicine, Neurological Clinic, University of Pisa, 56126 Pisa, Italy; 17Department of Clinical and Experimental Medicine, University of Messina, 98122 Messina, Italy; 18Department of Neurosciences, Biomedicine and Movement Sciences, Section of Clinical Neurology, University of Verona, 37129 Verona, Italy

**Keywords:** COL6-RM, *COL6A1*, *COL6A2*, *COL6A3*

## Abstract

Objective: To define the prevalence of variants in collagen VI genes through a next-generation sequencing (NGS) approach in undiagnosed patients with suspected neuromuscular disease and to propose a diagnostic flowchart to assess the real pathogenicity of those variants. Methods: In the past five years, we have collected clinical and molecular information on 512 patients with neuromuscular symptoms referred to our center. To pinpoint variants in COLVI genes and corroborate their real pathogenicity, we sketched a multistep flowchart, taking into consideration the bioinformatic weight of the gene variants, their correlation with clinical manifestations and possible effects on protein stability and expression. Results: In Step I, we identified variants in COLVI-related genes in 48 patients, of which three were homozygous variants (Group 1). Then, we sorted variants according to their CADD score, clinical data and complementary studies (such as muscle and skin biopsy, study of expression of COLVI on fibroblast or muscle and muscle magnetic resonance). We finally assessed how potentially pathogenic variants (two biallelic and 12 monoallelic) destabilize COL6A1-A2-A3 subunits. Overall, 15 out of 512 patients were prioritized according to this pipeline. In seven of them, we confirmed reduced or absent immunocytochemical expression of collagen VI in cultured skin fibroblasts or in muscle tissue. Conclusions: In a real-world diagnostic scenario applied to heterogeneous neuromuscular conditions, a multistep integration of clinical and molecular data allowed the identification of about 3% of those patients harboring pathogenetic collagen VI variants.

## 1. Introduction

Collagen VI-related myopathies (COL6-RM) are a group of rare genetic diseases with a heterogeneous phenotypic spectrum.

Within this mixed group, at least three distinct clinical entities can be identified: the most severe form, Ullrich’s congenital muscular dystrophy (UCMD; OMIM 254090), the mildest, Bethlem myopathy (BM; OMIM 158810), and intermediate phenotypes. However, common clinical characteristics can be recognized, such as muscle weakness, which mainly involves the muscles of the pelvic girdle, distal joints laxity, contractures of proximal joints (hips, knees, elbows, and spine), and skin abnormalities, such as pilar hyperkeratosis and keloid formation. Patients with the more severe form also have restrictive lung disease that can lead to respiratory failure [1,2]. The different forms present high allelic heterogeneity and reaching a full diagnosis is hard and laborious in most cases, also considering the different levels of penetrance of the gene variants.

Muscle biopsy in COL6-RM shows variable patterns ranging from nonspecific myopathic signs to dystrophic pictures. The reduction in collagen VI expression that could be studied on muscle tissue and on cultured skin fibroblasts is more informative. When available, muscle magnetic resonance imaging (MRI) could be of great help, as it can show a characteristic pattern on muscle involvement, particularly that of proximal muscles, the anterior muscular compartment of the thigh and the posterior muscular compartment of the leg. Two pathognomonic signs are also described in these patients: the “target” (“Central cloud”) sign in the rectus femoris muscle and the “sandwich” (“rim”, “outside-in”) sign in the vastus lateral muscle [3,4,5].

Diagnosis of COL6-RM is possible by detecting dominant or recessive mutations in the *COL6A1*, *COL6A2* and *COL6A3* genes and combining clinical and instrumental data. However, considering the large size of COLVI genes, until now, the diagnosis of COLVI-RM is underestimated and many patients are missed or incorrectly diagnosed.

On the other hand, the advent of new sequencing techniques has allowed the identification of many mutations whose pathogenicity remains uncertain or not always easy to corroborate.

The objectives of our study are manifold and include: the evaluation of the real prevalence of COLVI gene variants in neuromuscular patients undergoing routine genetic testing in third-level centers; defining the clinical or instrumental factors that may help in this process; and showing the percentage of patients in whom a definitive diagnosis can be reached. The importance of deepening our knowledge of COL6-RM is also made urgent in the face of emerging therapies and unprecedented opportunities to treat these patients [6,7].

## 2. Results

Figure 1 summarizes the flowchart used to define the real pathogenicity of variants, considering five different steps: Step I, defining the priority of variants in next-generation sequencing (NGS) analysis; Step II, based on the Combined Annotation Dependent Depletion (CADD) scores of variants observed in the gnomAD; Step III, collecting all available clinical data; Step IV, inspecting possible complementary studies (such as muscle and skin biopsy, study of expression of COLVI on fibroblast or muscle and muscle magnetic resonance); and Step V, based on predicted in silico protein stability changes (Figure 1).

Of the 512 samples subjected to Next-Generation Sequencing (NGS) analysis, we identified 48 cases harboring variants in the *COL6A1, COL6A2* and *COL6A3* genes: three variants were homozygous and 45 heterozygous. The remaining 464 patients were excluded from the study, since 59/512 had no COLVI gene variants and 405 had variants in other known genes, questioning the diagnosis of COL6-RM.

Overall, in the 48 cases, we identified 38 missense, one nonsense, six splice site, and three in/del leading to frameshift and predicting a prematurely truncated protein.

Among the 48 patients (26 males), 10 harbored variants in *COL6A1*, 17 carried variants in *COL6A2* (one homozygous) and 21 patients had variants in *COL6A3* (two homozygous) (Step I).

The CADD score of the different variants appeared distributed as in Figure 2: group 1 included three patients with a biallelic variant, of which two had CADD >25, whereas group 2 included 27 patients with a monoallelic variant, presenting a CADD score > 23. To increase the stringency, we decided to exclude patients whose missense had CADD <23 from this study (*n* = 18) (Figure 2).

Analyzing clinical data and complementary studies, we observed that all patients in group 1 presented a phenotype compatible with COL6-RM diagnosis. In two siblings (ID 1 and 2), symptoms presented at birth and were suggestive of UCMD. The two patients presented a typical weakness distribution with greater impairment of proximal muscles compared to distal ones and of lower limbs than upper ones. Both cases also presented rigid spine, ligament laxity, proximal contractures and skin alterations such as keratosis pilaris and velvety skin on fingers and toes. The younger brother required nocturnal non-invasive ventilation (NIV) and had never acquired independent walking. His older sister acquired autonomous walking, but she lost autonomous walking during her childhood. Only ID 1 underwent a muscle biopsy, which showed dystrophic signs and reduced COLVI expression. A reduction in COLVI expression was also found in her cultured skin fibroblasts. Muscle imaging in ID 1 showed marked and widespread fibro-adipose substitution of all muscles with a score of 3 at the level of the pelvis, paravertebral, intercostal and rectus abdominis muscles, according to the so-termed “Mercuri grading”. At the thigh level, its wide and marked involvement was evident, with a latero-mesial gravity gradient for the relative sparing of the adductors magnus and longus; the pathognomonic rim sign in the rectus femoris was also evident. In the leg muscles, there were a relative sparing of the tibialis anterior (Figure 3).

The clinical phenotype of patient ID 3 denoted specific muscle weakness affecting the pelvic girdle muscles and other typical signs of the COL6-RM phenotype, such as rigid spine, ligament laxity and keratosis pilaris. He did not undergo muscle and skin biopsies, but his muscle imaging showed features suggestive of a fibro-adipose infiltration of the pelvic (more evident at the level of the gluteal and tensor fascia lata muscles) and of the thigh muscles, with sparing of the sartorius, gracilis and intermediate portion of the long adductor muscles. In the leg, an adipose infiltration of the most anterior parts of the lateral gastrocnemius muscle on both sides was appreciated. These patterns are usually seen in COL6-RM.

Table 1 lists features in group 2 patients (*n* = 27). We performed a muscle biopsy in 21 patients, and we assessed COLVI expression in six. A skin biopsy was analyzed in seven patients, and MRI was done in 16 patients (Table 1). Global clinical evaluations permitted us to exclude six patients because their weakness distribution was not typical and suggested other neurological condition, i.e., in one case, signs of an asymmetrical involvement were reported (ID 17); in another case, there were only and exclusively signs of involvement of the face muscles (ID 22).

The remaining 21 patients had a normal clinical examination, or mild myalgias, or a weakness distribution typical of COL6-RM. Four patients (ID 12, 13, 16 and 20) presented only hyperCKemia (three or more times normal values), three presented only mild myalgia associated with either laxity (ID 6) or laxity and rigid spine (ID 19) or hyperCKemia (ID 24). Eleven patients (ID 7, 9, 10, 11, 18, 21, 25, 26, 27, 28, 30) presented weakness mainly in the pelvic girdle muscles or in the pelvic and scapular girdle muscles, of which eight had at least one additional sign associated with COLVI-RM. Three patients (ID 5, 14, 25) presented an involvement of the global musculature, of which two had at least another sign associated with COLVI-RM.

As for the 21 patients in group 2, we subsequently analyzed the result of the complementary examinations when they became available. Three patients (ID 7, 18, 30) presented dystrophic signs on muscle biopsy, of which two (ID 7, 18) also presented typical muscle involvement on muscle MRI and one (ID 18) a reduced expression of COLVI in a skin biopsy (Appendix A).

Six patients (ID 9, 10, 14, 16, 21, 28) presented myopathic signs on muscle biopsy. Of these, three patients (ID 9, 14, 16) had reduced expression of collagen VI in muscle, two (ID 14, 16) also had a reduced expression of COLVI in a skin biopsy, and two (ID 10, 28) had a normal level of expression of collagen VI in muscle while it was reduced in the skin (Appendix A). In a single case (ID 21), the expression of collagen VI was not evaluated.

Four patients (ID 14, 16, 21, 28) also underwent muscle MRI, which showed typical involvement in only two patients (ID 14 and 16).

Two patients had normal muscle biopsy and MRI (ID 19, 25), but one presented a reduction in collagen VI on skin biopsy (ID 25) and one had typical clinical involvement (ID19) (Appendix A).

One patient (ID 26) did not undergo a muscle biopsy, had normal MRI and presented a reduction in collagen VI on skin biopsy (Appendix A).

The remaining nine patients (ID 5, 6, 11, 12, 13, 20, 24, 27, 29) were excluded from further evaluations for the lack of data or for clinical and imaging data poorly matching with a COL6-RM phenotype in six cases (Step IV).

To further corroborate the pathogenicity of the missense variant, we studied the effect on the protein structure for seven missense variants. We were able to carry out this study only for variants in *COL6A1* and *COL6A2*, since the tridimensional structure of the COL6A3 protein has not yet been resolved.

All four *COL6A1* variants (c.842G>A/p.G281E, c.788G>A/p.G263D, c.787G>A/p.G263S and c.1315C>T/p.R439W detected in cases ID 10, 18, 19 and 25) determined a negative variation in folding free energy (ΔΔG) between wild-type and mutant structures, indicating a destabilizing effect (Figure 4a).

Three *COL6A2* variants (c.1806C>G/p.C602T, c.2785G>A/p.V929M, c.2991C>G/p.F997L detected in cases ID 14, 16 and 28, respectively) also determined a negative variation in folding free energy (ΔΔG), indicating a tendency of a destabilizing effect in one case (c.1806C>G/p.C602T found in ID 14) and a destabilizing effect in two (c.2785G>A/p.V929M, c.2991C>G/p.F997L found in ID 16 and 28, respectively) (Figure 4b).

## 3. Discussion

Neuromuscular pathologies are a large group of disorders including numerous rare forms presenting a broad spectrum of genetic and phenotypic variability. For this reason, a definitive diagnosis is not always easy to achieve [8,9,10,11,12].

Recent innovations in molecular genetics provide a fundamental help in the diagnostic process; however, this tool does not always provide easy-to-interpret results, and often the results do not allow a molecular diagnosis to be obtained.

An example is the heterogeneous group of COL6-RM. The COLVI genes, in particular *COL6A3*, are large and susceptible to genetic polymorphism, whose pathogenicity is not always easy to corroborate. This is also certainly due to the wide phenotypic heterogeneity that sometimes mimics other forms of myopathy.

To assess the real prevalence of COLVI variants in our study, we reassessed data from a five-year diagnostic work using NGS analysis in patients with symptoms or signs of suspected neuromuscular disease analyzed in multiple third-level Italian Neuromuscular disease (NMD) centers and investigated in a single laboratory.

The challenge of this work was to define a flowchart that could help to define the real pathogenicity of the COLVI variants (Figure 1). We found a 9.3% relative prevalence of variants in the *COL6A1*, *COL6A2* and *COL6A3* genes.

However, not all variants identified a pathological phenotype. Combining the strict bioinformatics scores based on ACMG grading and high-ranked CADD scores with the morphological and imaging data, we could attribute significance to most variants of uncertain significance. We cannot exclude that, in the present work, using a stringent phred-scaled CADD score, we only selected the “tip of the iceberg” and missed further COLVI involvement by dismissing hypomorphic variants potentially acting on the phenotype (e.g., group 3 patients). Our choice to select patients with mutations with a CADD > 23 was only an arbitrary choice; the mutations with a lower CADD must not be necessarily excluded in clinical practice, as they may in the future be associated with different phenotypes or represent modifiers of other myopathies. Furthermore, re-analysis of the available data in cases belonging to the latter group with class 3 mutations and CADD scores < 23 made a correlation with clinical, imaging and morphological features unlikely, unless they were minimal. However, we cannot weight in full these subtle effects, since no longitudinal evaluations were accessible to us in most of the cases.

The purpose of our study is to define a diagnostic approach comprehensive of genetic and bioinformatic analysis, clinical and imaging data and protein functional analysis. We want highlight that the outcomes of the above-mentioned analysis does not have the same weight in the diagnosis process: genetic analysis with family segregation and skin biopsy in association with bioinformatic analysis are the key driver for an accurate diagnosis.

Overall, in 7/15 cases, we could deepen our analyses of *COL6A1* and *COL6A2* variants with a functional approach to further comfort their pathogenetic role. Using Alpha-fold and multiple bioinformatic 3D tools, we could categorize 7 of the 14 variants on the stability of the protein. In all, we highlighted a destabilization for six (ID10 *COL6A1* c.842G> A: p.G281E; ID16 *COL6A2* c.2785G> A: p.V929M; ID18 *COL6A1* c.788G> A/p.G263D; ID19 *COL6A1* c.787G> A: p.G263S; ID25 *COL6A1* c.1315C> T: p.R439W; ID28 *COL6A2* c.2991C> G: p. F997L) and a tendency toward destabilization for a single variant (ID14 *COL6A2* c.1806C> G: p.C602T) (Figure 4a,b).

The focused collection of detailed information is critical in assessing the significance of multiple prediction tools. In five patients, we could directly collect their clinical progression, early weakness and imaging and morphological data. Patients ID1 and ID2 are siblings with the same variant, and the expression of collagen VI on the skin biopsy was reduced. Patients 3 and 9 presented a variant described as pathogenetic in the literature [8,13,14,15,16,17], and patient 9 (but also the suggestive patient 26) presented reduced collagen VI expression in cultured skin fibroblasts (Appendix A).

The remaining three patients (ID 7, 21 and 30) had a compatible clinic phenotype and an informative muscle biopsy. In multiple cases (*n* = 9), the examination of MRI and histology directly supported the grading obtained from genetic studies. This informativeness and correlation is particularly useful in cases such as patients ID12, ID13 and ID20, whose clinical data were normal and insufficient complementary investigations had been carried out. However, their variants had a high CADD score and in silico analyses showed a tendency of a destabilizing effect on the protein structure.

The identification by multiple-step analyses of about 3% of cases with potentially pathogenic variants appears to be in line with previous studies in the literature, although it is probably underestimated given the lack of complete data available for most cases [18,19,20]. In a Brazilian cohort, the definitive genetic diagnosis of COL6-RM was confirmed in 2.9% of patients [20]. In another large American cohort consisting of 4656 patients with suspected limb girdle muscle disease (LGMD), the pathogenicity of variants in the COLVI genes was corroborated in 3.1% of cases [19]. However, it should be noted that, in the latter cohort, all patients presented a strength deficit with a proximal-distal gradient, a phenotypic aspect also characteristic of COL6-RM and thus suggestive of a higher diagnostic rating [19]. On the other hand, in a small Spanish cohort, the percentage of mutations in the genes coding for collagen VI appeared slightly higher (4.8%) [18], though in that study, reduced collagen VI expression in the skin was available only in a few patients (4/10), and a compatible MRI was reported in a single case.

Our study added to the allelic and clinical heterogeneity of COL6-RM. We identified 12 new variants associated with COL6-RM phenotypes and supported the pathogenicity of two previously described variants. We also found a marked clinical variability of phenotypes that ranges from typical UCMD, to intermediate pictures, to BM, or oligosymptomatic individuals. While the identification of pictures compatible with UCMD appears straightforward, the distinction between BM and intermediate phenotypes is not well-defined. We also recorded atypical phenotypes, such as a patient with a completely silent neuromuscular examination who received gene testing for the identification of a moderate hyperCKemia (ID 16, Creatine phosphokinase-CPK levels were four times the norm). Of note, patient ID28 also presented a typical involvement of the proximal musculature associated with bilateral ptosis. A similar case with BM has been described before [21] in a patient harboring a mutation in *COL6A2*. In this patient, the expression of collagen VI on skin and muscle biopsy has not been evaluated and muscle MRI was not performed, limiting the significance of this variant in *COL6A2* and its association with ptosis and COL6-RM. However, our data appear to support the presence of ptosis in the COL6-RM phenotype, since we verified a lower expression of collagen VI on the skin of our patient. This specific case is instructive in that it is a further remark on how deep phenotyping is necessary when testing patients with neuromuscular disorders in routine laboratory screening. Furthermore, a double trouble cannot clearly be excluded; the patient might present variations in genes that we do not currently know, mutations that we may not have identified with NGS or modifying factors that we do not yet know. Ptosis, facial muscles and ophthalmoplegia/paresis, for example, were described in patients with an *Ryr1* mutation, although these symptoms were described as more frequent in recessive cases than in dominant/de novo cases [22].

In clinical practice, genotype–phenotype correlations would be useful to better attribute the pathogenicity of found variants. However, our study did not allow us to identify a clear correlation between genotype and phenotype, since the sample is too small.

## 4. Materials and Methods

From 2016 to 2021, DNA samples from 512 patients with symptoms or signs of suspected neuromuscular disease were collected and analyzed in our genetic laboratory at the IRCCS Stella Maris. The patients were referred from neurology, pediatric or neuropediatric units of 18 different Italian neuromuscular tertiary centers, as reported elsewhere (Neurol genet).

We used the SureSelect technology (Agilent, Santa Clara, CA, USA) and SureDesign software (earray.chem.agilent.com/suredesign/) to design a multiexon amplicon panel containing a total of 241 genes known to be associated with muscular dystrophies and myopathies. To analyze the data obtained from our study, we used a routine bioinformatic pipeline that adopts the QIAGEN Clinical Insight (QCI) Interpret analysis suite (https://apps.ingenuity.com, accessed on 13 December 2021, Qiagen, Venlo, The Netherlands). To assign pathogenicity, we used the following criteria: a sequence quality score greater than 30, a read depth greater than 30, and rare occurrence in publicly available polymorphic data sets (with a minor allele frequency <0.01% for autosomal dominant and <0.1% for autosomal recessive genes), with less than one occurrence in homozygosity in gnomADv2.1 (https://gnomad.broadinstitute.org/, latest access 15 December 2020) [23]. Each variant was also studied with different bioinformatics systems, including Varsome (https://varsome.com/, accessed on 13 December 2021) [24] and Combined Annotation Dependent Depletion (CADD, https://cadd.gs.washington.edu/snv) [25]. Putatively deleterious variants were validated by PCR-based standard capillary Sanger sequencing, both in patients and in relatives whose DNA was available for segregation studies, also to determine inheritance and phases of multiple gene variants and to establish whether variants had occurred de novo. Segregation in affected and unaffected relatives made it possible to better define pathogenic variants once we had identified those more likely to be disease-causative. We considered only patients with variants in COL6 genes, excluding those with certain variants in other genes. For COL6A1, we considered transcript NM_001848, for COL6A2 transcript NM_001849 and for COL6A3 transcript NM_004369.

The interpretation of the identified variants in COL6 genes is based on current knowledge and on the ACMG classification [26]. It has been shown that the optimal CADD phred-like score cut-off is between 20 and 25 [ref PMID: 30742610]. In this study, we used a stringent CADD score cut-off (>23) and Grantham score (≥151).

Each center was asked to fill in case report form (CRF) with clinical data (general and neurological examination, distribution of weakness, presence of other symptoms related to collagenopathies, age of onset of symptoms), anamnestic data (pathological history with regard to other comorbid pathologies and possible cardiac involvement, family history), muscle biopsy report, muscle and fibroblast protein expression, blood creatine kinase assay, electromyography and MRI results of identified cases, if available.

Finally, we studied the effect of some variants on protein structure. To predict protein stability changes upon mutation, in terms of variation in folding free energy (ΔΔG) between wild-type and mutant structures, we employed nine different computational methods: DynaMut2 (PMID 32881105), ENCoM (PMID 24762569), mCSM (PMID 24281696), SDM (PMID 28525590), DUET (PMID 24829462), MUpro (PMID 16372356), CUPSAT (PMID 16845001), MAESTRO (PMID 25885774) and PremPS (PMID 33378330). Since no solved 3D structures are available for COL6A1 and COL6A2 proteins in the Protein Data Bank (https://www.rcsb.org/), the predicted 3D protein structures were sourced from AlphaFold Protein Structure Database (https://alphafold.ebi.ac.uk/, PMID 34265844). 3D structures of *COL6A3* are not available in any database.

## 5. Conclusions

To the best of our knowledge, this is the first study that systematically used a multiple integrated level of information to prioritize variants in COL6-RM genes in routine diagnostic practice. This study underlines the need to accurately collect clinical data and complementary examinations (such as muscle and skin biopsy, study of expression of COLVI on fibroblast or muscle and muscle magnetic resonance) and to combine such data at multiple levels through shared registers.

Although with several limitations, including lack of complete complementary studies available to all patients, poor segregation studies in the families, and the absence of protein studies in silico or in vitro for most cases, our study contributed to define about 3% of causative patients in real-world NGS-based diagnosis, an information critical when new therapeutic opportunities are able to halt or limit the muscular damage in COL6-RM.

## Figures and Tables

**Figure 1 ijms-23-14567-f001:**
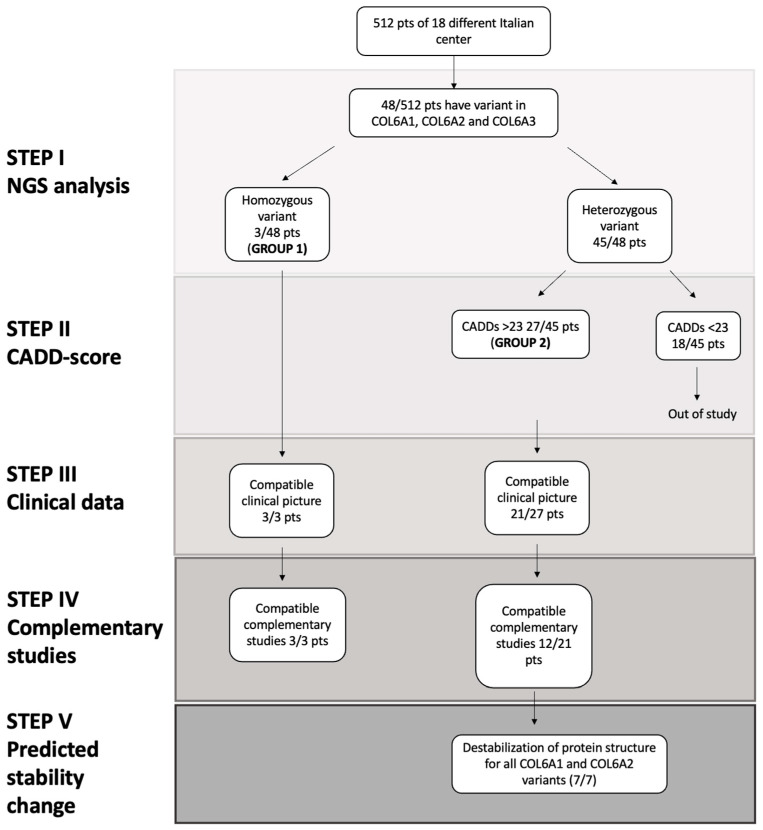
Diagnostic flowchart divided into five progressive steps: Step I Next-generation sequencing (NGS) analysis; Step II CADD score; Step III clinical data; Step IV complementary studies (such as muscle and skin biopsy, study of expression of COLVI on fibroblast or muscle and muscle magnetic resonance); and Step V predicted stability change.

**Figure 2 ijms-23-14567-f002:**
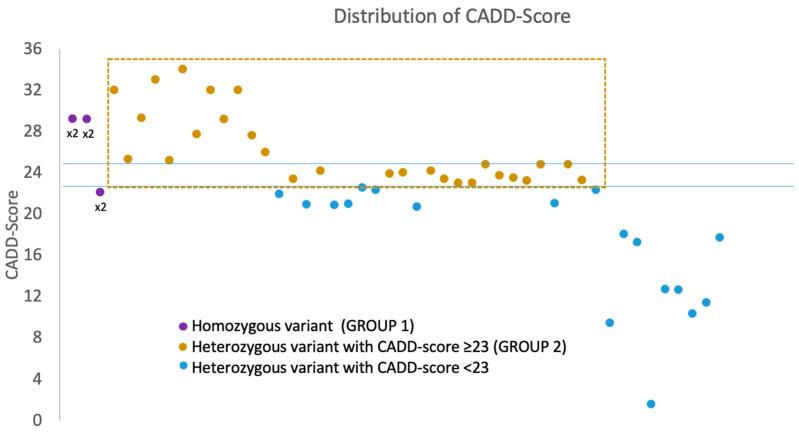
Graph showing the dispersion of variants (*x*-axis) according to the CADD score (*y*-axis). The points in purple represent the homozygous variants, the points in ocher the heterozygous variants with a CADD score greater than 23 (Group 2) and the points in blue the heterozygous variants with a CADD score less than 23 not covered by our study.

**Figure 3 ijms-23-14567-f003:**
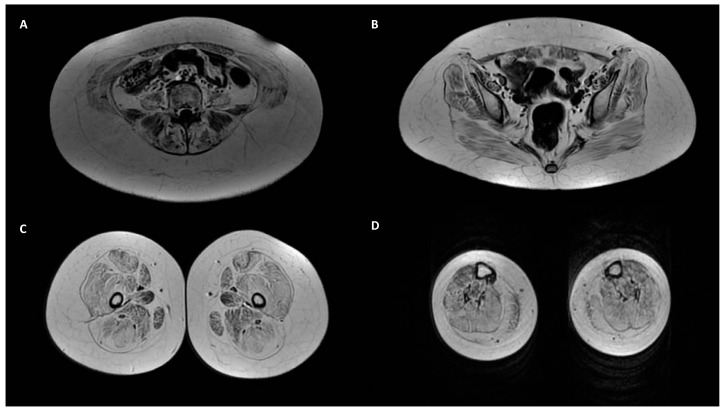
Axial T1 SE imaging of patient ID 1 that shows: marked and widespread fibro-adipose substitution of muscles (grading 3/4 according to “Mercuri score”) at the level of the pelvis (**A**,**B**); wide and marked involvement with a latero-mesial gravity gradient of thighs (**C**); wide and marked involvement of leg muscles (**D**).

**Figure 4 ijms-23-14567-f004:**
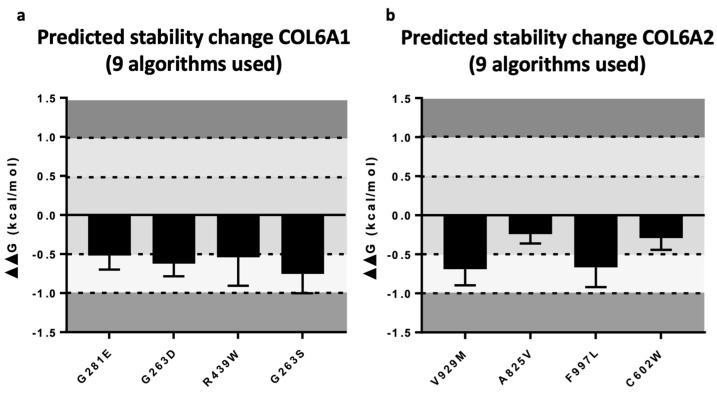
Representation of the prediction change in stability of subunits A1 (**a**) and A2 (**b**) of COLVI due to the missense identified in this study. Variation in folding free energy (∆∆G) is predictors of protein stability changes upon single-point variations: ΔΔG > 1.0 indicate a probable stabilizing effect; 0.5 < ΔΔG < 1.0 indicate a likely stabilizing effect; −0.5 < ΔΔG < 0.5 indicate no significant change in stability; −1.0 < ΔΔG < −0.5 indicate a likely destabilizing effect; ΔΔG < −1.0 indicate a probable destabilizing effect.

**Table 1 ijms-23-14567-t001:** Description of phenotype and genotype of 30 selected patients (three with homozygous variants and 27 with CADD score variants above 23). Het = heterozygous, wt = wild-type, x= present.

ID	Age Onset	Gender	Gene	Mutation	CADD-Score	Varsome	Muscle Weakness	Joint Laxity	Contracture or Retraction of Achilleus Tendon	Rigid Spine Scoliosis Hip Alt.	Skin Sign	Lung Alt.	CK Elevation	Muscle Biopsy and COLVI Expression	Skin Biopsy and COLVI Expression	Muscle Magnetic Resonance	Segregation
1	Congenital	F	COL6A3	c.7176delG: p.G2392fs* **homo**	29.2	Pathogenic	Global (>shoulder and pelvic girdle)	x	x		x		mild	Myopathic sign; COLVI reduction	Reduction	Typical	Father and mother het
2	Congenital	M	COL6A3	c.7176delG: p.G2392fs* **homo**	29.2	Pathogenic	Global (>shoulder and pelvic girdle)	x	x		x	x	mild	N/A	N/A	N/A	Father and mother het
3	14	M	COL6A2	c.1970-9G>A **homo**	22.1	Likely pathogenic	Pelvic girdle	x		x	x		Normal	N/A	N/A	Typical	Father and mother het
4	58	M	COL6A1	c.3013C>T: p.R1005C	32	Uncertain significance	Distal muscles of leg						Normal	Myopathic sign	N/A	Normal	N/A
5	60	M	COL6A2	c.2474C>T: p.A825V	25.3	Uncertain significance	Global						Normal	Myopathic sign; COLVI normal	N/A	Normal	N/A
6	N/A	F	COL6A2	c.791G>A: p.R264H	29.3	Likely Pathogenic	Myalgia, generalized asthenia	x					Normal	N/A	N/A	Atypical	N/A
7	44	M	COL6A2	c.2461+1G>A	33	Pathogenic	Shoulder and pelvic girdle (>pelvic)						N/A	Dystrophic signs	N/A	Typical	N/A
8	65	F	COL6A1	c.3006C>A: p.H1002Q	25.2	Uncertain significance	Distal muscles of arm and leg						N/A	N/A	N/A	N/A	N/A
9	Congenital	F	COL6A3	c.6210+1G>A	34	Pathogenetic	Global (>shoulder and pelvic girdle)			x			mild	Myopathic sign; COLVI reduction	Normal	N/A	N/A
10	Congenital	M	COL6A1	c.842G>A: p.G281E	27.7	Likely pathogenic	Global (>shoulder and pelvic girdle)	x	x	x			mild	Myopathic sign; COLVI normal	Intracellular distribution	N/A	N/A
11	40	M	COL6A3	c.2029C>T: p.R677C	32	Uncertain significance	Trunk, shoulder and pelvic girdle						severe	Atypical sign	N/A	Atypical	N/A
12	13	F	COL6A2	c.1358G>A: p.R453H	29.2	Likely Pathogenic	Normal						Episodic	N/A	N/A	N/A	N/A
13	N/A	M	COL6A2	c.1395+2T>C	32	Pathogenic	Normal						severe	N/A	N/A	N/A	N/A
14	Congenital	F	COL6A2	c.1806C>G: p.C602T	27.6	Uncertain Significance	Global	x	x	x			normal	Myopathic sign; COLVI reduction	Reduction	Typical	Father het
15	42	F	COL6A3	c.4121A>T: p.D1374V	26	Uncertain significance	Distal muscles of leg						low	Myopathic sign	N/A	N/A	N/A
16	17	M	COL6A2	c.2785G>A: p.V929M	23.4	Uncertain Significance	Normal						moderate	Myopathic sign; COLVI reduction	Reduction	Typical	N/A
17	75	M	COL6A3	c.6224C>T: p.P2075L	24.2	Uncertain Significance	Right shoulder girdle and orbicular muscles						severe	Atypical sign	N/A	N/A	N/A
18	Congenital	F	COL6A1	c.788G>A: p.G263D	23.9	Likely pathogenic	Global (>shoulder and pelvic girdle)		x				mild	Dystrophic signs	Reduction	Typical	Parents wt
19	6	M	COL6A1	c.787G>A: p.G263S	24	Likely pathogenic	Myalgia	x		x			severe	Normal	N/A	Normal	N/A
20	4	F	COL6A3	c.2845G>A: p.A949T	24.2	Uncertain Significance	Normal						moderate	Myopathic sign	N/A	N/A	N/A
21	Congenital	F	COL6A3	c.8359G>A: p.A2787T	23.4	Uncertain Significance	Shoulder and pelvic girdle	x		x		x	normal	Myopathic sign	N/A	Normal	N/A
22	4	F	COL6A3	c.8009C>T: p.A2670V	23	Likely pathogenetic	Orbicular muscles						normal	Atypical sign	N/A	Normal	N/A
23	60	F	COL6A1	c.2635A>G: p.S879G	23	Uncertain Significance	Trunk, shoulder girdle and facial muscles						moderate	Myopathic sign	N/A	N/A	N/A
24	40	F	COL6A2	c.2182G>A: p.V728M	24.8	Uncertain Significance	Myalgia						severe	Atypical sign	N/A	Normal	N/A
25	Congenital	F	COL6A1	c.1315C>T: p.R439W	23.7	Likely pathogenic	Trunk, shoulder and pelvic girdle		x	x		x	Normal	Normal	Reduction	Normal	Mother het
26	Congenital	F	COL6A3	c.787G>A: p.D263N	23.5	Uncertain Significance	Pelvic girdle	x		x	x		Normal	N/A	Reduction	Normal	Father het
27	Congenital	M	COL6A2	c.2950G>A: p.V984M	23.2	Uncertain Significance	Pelvic girdle	x					Normal	N/A	Normal	Atypical	Father het
28	11	M	COL6A2	c.2991C>G: p.F997L	24.8	Uncertain Significance	Pelvic girdle (ptosis)						Mild	Myopathic sign; COLVI normal	Reduction	Normal	Parents wt
29	30	M	COL6A3	c.2212A>T: p.R738W	24.8	Uncertain Significance	Global (ptosis)		x				Mild	Inflammatory signs	N/A	Atypical	N/A
30	40	M	COL6A3	c.7258C>T: p.R2420W	23.3	Uncertain Significance	Shoulder and pelvic girdle		x	x			Severe	Dystrophic signs	N/A	N/A	N/A

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
