# Peer review of "A Schematic Approach to Defining the Prevalence of COL VI Variants in Five Years of Next-Generation Sequencing"

_ijms, 2022, doi:10.3390/ijms232314567_

Round 1
Reviewer 1 Report
The MS describes the diagnostic performance in the setting of a network of 18 Italian reference centres for identifying disease causative mutations in Collagen VI genes in patients with a suspected myopathy or muscle disease.
Minor comment
I would suggest choosing another term for instrumental data which here refers to complementary studies such as muscle and skin biopsy, maybe supporting data or biochemical/morphological data through the MS.
Major comments
It is understandable that the workflow is somehow strict and that possible "outliers" are removed. However, the work would benefit in terms of clarity if some of this "filters" or thresholds are explained a bit better. For example, The authors say they exclude 6 patients because the distribution of the weakness was not typical but in the following paragraph they say that out of these 21 patients, some only had myalgia and others no clinical signs, why were those not excluded as well?.
It would also be interesting to now if the authors are planning to investigate further the mutations with CADD score < 23, there may be acting through unexpected mechanisms or may represent disease modifiers. This is partly covered in the discussion.
According to Materials and Methods segregation studies were performed in order to determine mode of inheritance or de novo occurrence. However, these are not reported for individual variants or in the main table. This should be included.
Regarding the patient with ptosis and reduced collagen VI in skin and muscle. Is it possible that this reduction may be secondary and that the primary cause of the ptosis may be other?. For example, although it may not have been reported in the literature, collagen VI deficiency shares some features with RYR1 related congenital myopathy and a secondary reduction in collagen VI can be found in fibroblasts in patients with RYR1 mutations. This possibility should be discussed, at least mention that there may be a secondary reduction in collagen VI due to yet unknown interactions with other proteins as it occurs with other CMDs or congenital myopathies.
Author Response
We want to thank the reviewers for the expert comments and the opportunity to improve our manuscript.
A revised manuscript has been prepared taking into account all the comments of the expert reviewers (highlighted in the text) and a point-by-point response to reviewers has been prepared. The text has been edited for spelling and typos and the accuracy of the references has been re-checked, verified, and renumbered if necessary.
Dear revisor 1
Concerning Minor comment:
I would suggest choosing another term for instrumental data which here refers to complementary studies such as muscle and skin biopsy, maybe supporting data or biochemical/morphological data through the MS.
Answer: Wr thank the expert referee for this comment. In the revised text, we modified the term “instrumental data” in the manuscript with the term “complementary studies” or “complementary investigations/examinations” (see page 1 ln 40; page 2 ln 92; page 3 ln 97; page 4 ln 120; page 8 ln 174; page 10 ln 269; page 11 in 361 and 365; and Figure 1).
Major comments
It is understandable that the workflow is somehow strict and that possible "outliers" are removed. However, the work would benefit in terms of clarity if some of this "filters" or thresholds are explained a bit better. For example, The authors say they exclude 6 patients because the distribution of the weakness was not typical but in the following paragraph they say that out of these 21 patients, some only had myalgia and others no clinical signs, why were those not excluded as well?.
Answer: We appreciated this specific comment that offers the opportunity to clarify our study. To avoid exclusion of outliers in clinical phenotypes, we excluded cases with non-neuromuscular presentation, even if minimal or subtle. For example, in one case signs of an asymmetrical involvement were reported (ID 17), in another case there were only and exclusively signs of involvement of facial muscles (ID 22).We tried to make our message more clear and exhaustive (see page 5 ln 155).
It would also be interesting to know if the authors are planning to investigate further the mutations with CADD score < 23, there may be acting through unexpected mechanisms or may represent disease modifiers. This is partly covered in the discussion.
Answer: Thanks for this suggestion, we have better argued this information in the discussion (see page 9 ln 237), underlining that our choice to select patients with mutations with CADD> 23 was an arbitrary choice to gather a more stringent analysis of COL VI-related myopathies and that the mutations with lower CADD (with scores between 20 and 23) have not to be fully excluded in clinical practice, pending more precise evaluation of atypical phenotypes or role as modifiers of other myopathies. It would be useful in the future to be adopt more complex statistical tools to weigh the pathogenetic role of the multiple variants emerging from NGS in defining and modifying the different clinical phenotypes.
According to Materials and Methods segregation studies were performed in order to determine mode of inheritance or de novo occurrence. However, these are not reported for individual variants or in the main table. This should be included.
Answer: Thanks for this suggestion, We included data in Table 1 whenever available.
Regarding the patient with ptosis and reduced collagen VI in skin and muscle. Is it possible that this reduction may be secondary and that the primary cause of the ptosis may be other?. For example, although it may not have been reported in the literature, collagen VI deficiency shares some features with RYR1 related congenital myopathy and a secondary reduction in collagen VI can be found in fibroblasts in patients with RYR1 mutations. This possibility should be discussed, at least mention that there may be a secondary reduction in collagen VI due to yet unknown interactions with other proteins as it occurs with other CMDs or congenital myopathies.
Answer: We acknowledge that a double trouble condition cannot be excluded a priori. In our case no RYR1 mutations were found. (see page 10 ln 302).
We believe that in this new form our manuscript adds original information in COL6-RD solving the problem of the pathogenicity of genetic variants in genes causing Collagen VI-related disorders in real-world diagnostic settings and replying satisfactorily to the concerns raised by the expert reviewers.
We hope that you will find at this stage our manuscript potentially suitable for publication.
Sincerely,
Guja Astrea
e-mail:guja.astrea@fsm.unipi.it
Reviewer 2 Report
This manuscript contributes to solving the problem of the pathogenicity of genetic variants in genes causing Collagen VI-related myopathies (COL6-RM) in real-world diagnostic settings. COL6-RM genes have a high frequency of genetic variants, and the spectrum of clinical manifestation is wide and heterogeneous. Hence, the interpretation of genetic testing results proved not to be straightforward.
The authors propose a workflow combining various bioinformatic analyses, clinical and instrumental data. The workflow helped them to solve diagnostics in 11 out of 30 patients.
The manuscript is written in a concise and clear manner.
A major suggestion is to specify instrumental data more precisely. Particularly, it will be nice to elaborate usefulness of measuring gene expression levels in skin biopsy, genetic analysis of family members, and MRI. Is there any priority in the order of the above-mentioned analysis to increase diagnostic yield? Whether skin biopsy should replace muscle biopsy when the genetic analysis is already done? Whether the bioinformatic analysis of protein can replace skin biopsy?
Minor suggestions:
- Table 2 is only a continuation of Table 1 and its title should be Table 1, continued, for example.
- Please, add an additional column to Table 1 with data about whether mutation arose de novo or is inherited, as well as whether the segregationally analysis was performed.
- All Figure and Table legends should be more self-explanatory.
Author Response
We want to thank the reviewers for the expert comments and the opportunity to improve our manuscript.
A revised manuscript has been prepared taking into account all the comments of the expert reviewers (highlighted in the text) and a point-by-point response to reviewers has been prepared. The text has been edited for spelling and typos and the accuracy of the references has been re-checked, verified, and renumbered if necessary.
Dear revisor 2
Major suggestions: a major suggestion is to specify instrumental data more precisely. Particularly, it will be nice to elaborate usefulness of measuring gene expression levels in skin biopsy, genetic analysis of family members, and MRI. Is there any priority in the order of the above-mentioned analysis to increase diagnostic yield? Whether skin biopsy should replace muscle biopsy when the genetic analysis is already done? Whether the bioinformatic analysis of protein can replace skin biopsy?
Answer: Thanks for the tip. A limitation of our study is given precisely by the fact that the patients derived from multiple centers and clinical information were not always possible in terms of morphological data and segregation in the family. Also, muscle MRI was not standard test in many centers at the collection of these patients. We agree with the reviewer that a more complete clinical and imaging study combined with immunohistochemical investigations in skin fibroblasts can facilitate prioritize and evaluate variants in collagen VI genes. This important suggestion that allowed us to better explain these topics and specify the different weight of complementary study in the diagnosis process (see paragraph discussion line 245 page 9).
Minor suggestions:
- Table 2 is only a continuation of Table 1 and its title should be Table 1, continued, for example.
Answer: We modified the table as suggested.
- Please, add an additional column to Table 1 with data about whether mutation arose de novo or is inherited, as well as whether the segregationally analysis was performed.
Answer: We modified the table as suggested.
- All Figure and Table legends should be more self-explanatory.
Answer: We modified the figures and tables to make them more readable.
We believe that in this new form our manuscript adds original information in COL6-RD solving the problem of the pathogenicity of genetic variants in genes causing Collagen VI-related disorders in real-world diagnostic settings and replying satisfactorily to the concerns raised by the expert reviewers.
We hope that you will find at this stage our manuscript potentially suitable for publication.
Sincerely,
Guja Astrea
e-mail:guja.astrea@fsm.unipi.it
Round 2
Reviewer 1 Report
The authors have replied to my comments-suggestions and the MS is acceptable for publication